# Probabilistic Spatial Transformer Networks

**Pola Schwöbel**[1]      **Frederik Warburg**[1]      **Martin Jørgensen**[2]      **Kristoffer H. Madsen**[1,3]      **Søren Hauberg**[1]

[1] Section for Cognitive Systems, DTU Compute, Technical University of Denmark, Copenhagen, Denmark
[2] Machine Learning Research Group, Department of Engineering Science, University of Oxford, Oxford, UK
[3] Danish Research Centre for Magnetic Resonance, Centre for Functional and Diagnostic Imaging and Research, Copenhagen University Hospital Hvidovre, Hvidovre, Denmark

## Abstract

Spatial Transformer Networks (STNs) estimate image transformations that can improve downstream tasks by 'zooming in' on relevant regions in an image. However, STNs are hard to train and sensitive to mis-predictions of transformations. To circumvent these limitations, we propose a probabilistic extension that estimates a stochastic transformation rather than a deterministic one. Marginalizing transformations allows us to consider each image at multiple poses, which makes the localization task easier and the training more robust. As an additional benefit, the stochastic transformations act as a localized, learned data augmentation that improves the downstream tasks. We show across standard imaging benchmarks and on a challenging real-world dataset that these two properties lead to improved classification performance, robustness and model calibration. We further demonstrate that the approach generalizes to non-visual domains by improving model performance on time-series data.

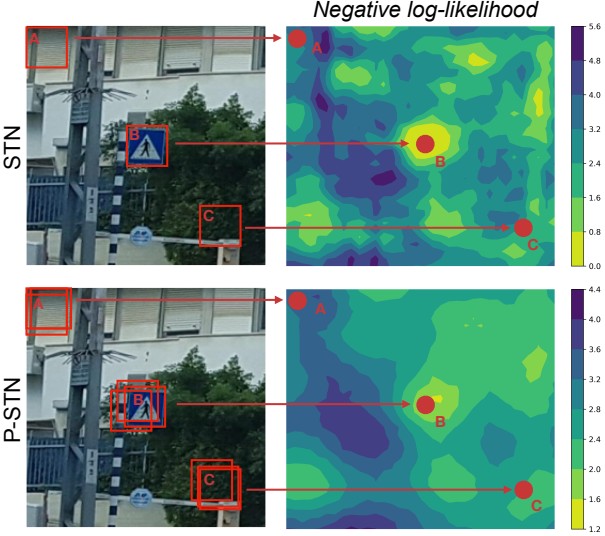

Figure 1: The Probabilistic Spatial Transformer Network (P-STN) marginalizes over a distribution of possible input transformations. By 'looking in multiple places' we hope to stabilize the brittle nature of the regular spatial transformer: The P-STN loss landscape is significantly more smooth and with fewer local minima compared to the STN.

## 1   INTRODUCTION

The *Spatial Transformer Network (STN)* [Jaderberg et al., 2015] predicts a *transformation* on input data in order to simplify a downstream task. For example, a neural network might benefit from e.g. 'zooming in' on relevant parts of an image, remove unwarranted image rotations, or time-normalize sequence data before making predictions. In principle, this can improve robustness, interpretability and efficiency of the model. However, in practice, the situation is not as ideal. Both at training and test time, the STN is sensitive to small mis-predictions of transformations. For example, if the STN zooms in on the wrong part of an image, then the signal is lost for the downstream task, e.g. see crop A and C in Fig. 1. The empirical impact is that STNs are

difficult to train and often do not live up to their promise.

From a probabilistic perspective, this sensitivity has an obvious solution: we should estimate the posterior over the applied transformation and marginalize accordingly. This amounts to 'trying many different transformations', and should improve robustness. It is exactly this approach we investigate.

STNs consist of two parts. A localization network performs the transformation task, i.e. it estimates the transformation parameters $\theta$ for a given image $I$ and applies the corresponding transformation $T_\theta(I)$. A standard neural network performs the downstream task on the transformed image, i.e. computing $p(y|T_\theta(I))$. Since we are concerned with classi-

*Accepted for the 38th Conference on Uncertainty in Artificial Intelligence* (UAI 2022).

fication tasks, we will refer to the latter as the classifier, but note that the approach generalizes to other tasks.

In our probabilistic STN (P-STN), we estimate a distribution over transformations that we marginalize: $p(y|I) = \int p(y|T_\theta(I))d\theta$. We approximate this intractable integral via Monte Carlo, i.e. we sample transformations. These transformation samples produce different transformed versions of the input image, $\{T_\theta^s(I)\}_{s=1\ldots S}$. The classifier makes predictions on all samples, and we aggregate the predictions. Figure 2 shows the model architecture.

We hypothesize that marginalizing image transformation has benefits for both parts of the model. For the *localization* network, our model gets to 'try many different transformations' through random sampling. This should improve the localization. Secondly, the classifier now gets presented with different transformed versions of the input image through Monte Carlo samples $\{T_\theta^s(I)\}_{s=1\ldots S}$. Interestingly, this corresponds to a type of data augmentation, which should improve classification.

We verify these hypotheses by making the following contributions:

1. We develop the Probabilistic Spatial Transformer; a hierarchical Bayesian model over image transformations.

2. We perform variational inference to fit the transformation model as well as downstream model end-to-end, using only label information.

3. We experimentally demonstrate that our model achieves better localization, increased classification accuracy (resulting from learned per-image data augmentation) and improved calibration.

## 2 RELATED WORK

**Spatial transformer networks** apply a spatial transformation to the input data as part of an end-to-end trained model [Jaderberg et al., 2015]. The transformation parameters are estimated from each input separately through a neural network. Most commonly, STNs implement simple affine transformations, such that the network can learn to zoom in on relevant parts of an image before solving the task at hand. STNs have shown themselves to be useful for both generative and discriminative tasks, and have seen applications to different data modalities [Jaderberg et al., 2015, Detlefsen and Hauberg, 2019, Detlefsen et al., 2018, Shapira Weber et al., 2019, Sønderby et al., 2015, Lin and Lucey, 2016, Kanazawa et al., 2016]. We propose a probabilistic extension of this idea, replacing the usual likelihood maximization with marginalization over transformations.

**Bayesian deep learning** aims to solve probabilistic computations in deep neural networks. Priors are put on weights and marginalized at training and test time, often yielding useful uncertainties in the posterior predictive. The required computations are in general intractable, and approaches differ mainly in the type of approximation to the weight posterior. Gal and Ghahramani [2016] propose to view dropout as a Bernoulli approximation to the weight posterior (i.e. randomly switching each weight on or off). The Laplace approximation [MacKay, 1992, Daxberger et al., 2021] places a Gaussian posterior over a trained neural network's weights. Another generally successful way to obtain predictive uncertainties is to simply train an ensemble of models. Originally proposed as an alternative to Bayesian DL [Lakshminarayanan et al., 2017], the approach can be interpreted in the Bayesian framework by interpreting the weights of the trained ensemble members as samples from a weight posterior [Gustafsson et al., 2020]. Similar to our method, Blundell et al. [2015] choose a variational approach with a simple Gaussian mean field posterior over weights. Our approach differs from standard Bayesian DL in that we are not reasoning about distributions over neural network weights $p(w)$, but instead a subnetwork's (i.e. the localizer's) *outputs* $p(\theta)$. Drawing from the posterior over image transformations, we effectively recover data augmentation.

**Data augmentation (DA)** is a useful way to increase the amount of available data [LeCun et al., 1995, Krizhevsky et al., 2012]. DA requires prior knowledge about the structure of the data: the target $y$ is assumed to be invariant to certain transformations of the observation $I$. Invariance assumptions are usually straight forward for natural images. Thus, DA is common for image data, where the transformation family is often chosen to be rotations, scalings, and similar [Goodfellow et al., 2009, Baird, 1992, Simard et al., 2003, Krizhevsky et al., 2012, Loosli et al., 2007]. The general trend is that, beyond 'intuitive' data such as images, gathering an invariance prior is difficult, and DA is often hard to realize through manual tuning.

**Learned data augmentation** provides a more principled approach to artificially extending datasets. Hauberg et al. [2016] estimate an augmentation scheme from the training data via pre-aligning images in an unsupervised manner. The approach allows for significantly more complex transformations than the usual affine family, but the unsupervised nature and the implied two-step training process render the approach suboptimal. Similarly, Cubuk et al. [2019, 2020] use reinforcement learning and grid search to learn data augmentation schemes, but rely on validation data rather than an end-to-end formulation.

Learning data augmentation end-to-end requires a loss function suitable for model selection, as we are effectively trying to learn an inductive bias. Based on this realization, Van der Wilk et al. [2018] learn DA end-to-end in Gaussian processes (GPs) via the marginal likelihood, a suitable loss for model selection and thus invariance learning [MacKay, 2003]. The marginal likelihood is hard to compute for NNs, so Schwöbel et al. [2022] extend this idea to NNs by consid-

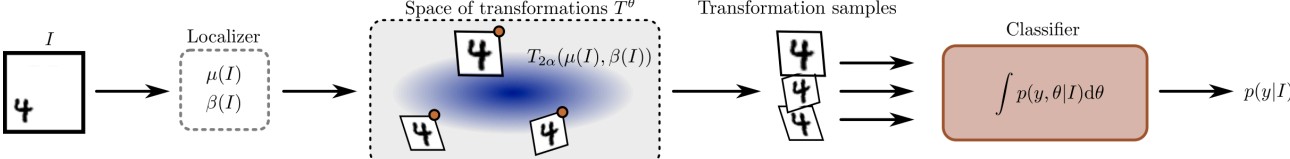

Figure 2: The P-STN pipeline. From the observed image $I$, a distribution of transformations is estimated. Samples from this distribution are applied to the observed image to produce augmented samples, which are fed to a classifier that averages across samples. In the deterministic STN case, the localizer only computes one transformation $\theta(I)$, which can be thought of as the maximum likelihood solution. Instead of the multiple transformation samples, we obtain a single $T_\theta(I)$ in this case.

ering a deep kernel model, i.e. a neural network with a GP in the last layer. Benton et al. [2020] instead use the standard, maximum likelihood loss and explicitly regularize towards non-zero augmentations. Our model differs from existing data augmentation approaches — learned and non-learned — in that we estimate local, i.e. *per-image* transformations instead of a global augmentation scheme.

## 3   BACKGROUND

The STN localiser module estimates a transformation $\theta(x)$ that transforms a coordinate grid and interpolates an image accordingly. The classifier module takes the transformed image and computes $p(y|T_\theta(x))$. Both the localizer and classifier are neural networks. The STN can be trained end-to-end with only label information as long as the image transformations are parameterized in a differentiable manner.

**Affine transformations** are a simple class of transformations that can be differentiably parameterized. We limit ourselves to the subset of affine transformations containing rotation, isotropic scaling and translation in $x$ and $y$. In two dimensions (and the corresponding three-dimensional homogeneous coordinates), we thus learn $\theta = (r, s, t_x, t_y)$ which parameterizes the affine matrix

$$A_\theta = \begin{bmatrix} s \cdot \cos r & -s \cdot \sin r & t_x \\ s \cdot \sin r & s \cdot \cos r & t_y \\ 0 & 0 & 1 \end{bmatrix} \in \mathbb{R}^{3 \times 3}, \ s > 0. \quad (1)$$

Since $\det(A_\theta) = s^2$, the constraint $s > 0$ ensures invertibility and can be implemented as seen in Detlefsen et al. [2018]. In practice, the STN estimates well-behaved, non-collapsing transformations without implementing the constraint explicitly. $T_\theta(I)$ is applied by transforming a grid of the target image size by $A_\theta$ and interpolating the source image at the resulting coordinates (see Jaderberg et al. [2015] for details).

**Diffeomorphic transformations** (i.e. transformations that are differentiable, invertible and possess a differentiable inverse) are more general than affine transformations, and are not limited to the spatial domain. Freifeld et al. [2017] construct diffeomorphisms from continuous piecewise-affine velocity fields as follows. The transformation domain $\Omega$ is

divided into subsets and an affine matrix is defined on each cell $c$ of such a tessellation. Each affine matrix $A_{\theta_c}$ induces a vector field mapping each point $x \in c$ to a new position $v^{\theta_c} : x \mapsto A_{\theta_c} x$. These velocity fields are then integrated to form a trajectory for each image point $x$

$$\phi^\theta(x; 1) = x + \int_0^1 v^\theta(\phi(x; \tau)) \mathrm{d}\tau.$$

Given boundary and invertibility constraints [Freifeld et al., 2017], such a collection of affine matrices $\{A_{\theta_c}\}_{c \subset \Omega}$ defines a diffeomorphic transformation $T^\theta : x \mapsto \phi^\theta(x, 1)$.

The libcpab library [Detlefsen, 2018] provides an efficient implementation for this approach, specifically optimized for use in a deep learning context where fast gradient evaluations are crucial. The author successfully employs CPAB-transformations within a Spatial Transformer Network [Detlefsen et al., 2018].

## 4   PROBABILISTIC SPATIAL TRANSFORMER NETWORK

The P-STN is a probabilistic extension of the STN, where we replace the deterministic transformation $\theta(I)$ with a posterior over transformations $p(\theta|I)$. Figure 2 illustrates the proposed pipeline. We assume observed data of the form $\mathcal{D} = \{y_i, I_i\}_{i=1}^N$, where $y$ is the target variable (e.g. class label), and $I$ are observations of the covariates. For presentation purposes, we will consider the latter to be images, but the approach applies to any spatio-temporal data.

### 4.1   THE MODEL

Recall that STNs are trained end-to-end for the downstream task using only label information. Thus, while we observe $y$, $\theta$ is a latent variable. We model it to be governed by a second latent variable $\lambda$. $\lambda$ is a precision parameter, effectively stopping the localization distribution (i.e. the amount of 'data augmentation' we introduce) from collapsing. The necessity for non-collapsing augmentation is discussed in Benton et al. [2020], Van der Wilk et al. [2018] and Schwöbel et al. [2022].

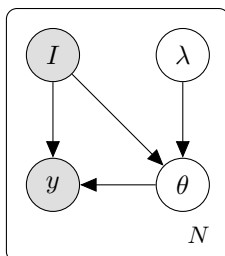

Figure 3: A graphical representation of the model structure. Grey nodes are observables and white are latents.

We wish to infer the latent variables in a Bayesian manner. This entails computing the (log-)marginal likelihood of the observed

$$\log p(I, y) = \log \iint p(I, y, \theta, \lambda) \mathrm{d}\theta \mathrm{d}\lambda. \tag{2}$$

We let the joint distribution factorize as (see Fig. 3)

$$p(y, I, \theta, \lambda) = p(y|I, \theta, \lambda) p(I, \theta, \lambda) \tag{3}$$

$$= p(y|I, \theta) p(\theta|\lambda, I) p(\lambda) p(I). \tag{4}$$

Notice $p(I)$ is unaffected by model parameters $\lambda$ and $\theta$, and in this sense can be specified without affecting the model. The distribution over $\theta$ depends on observed covariates in the following way

$$p(\theta|\lambda, I) = \mathcal{N}(\theta|\mu(I), 1/\lambda), \tag{5}$$

where $\mu(I)$ is a function parametrised by a neural network, i.e. $\mu(I) := \mu_\Phi(I)$ for model parameters $\Phi$. The prior over $\lambda$ is a Gamma distribution, i.e.

$$p(\lambda_i) = \Gamma(\alpha_0, \beta_0). \tag{6}$$

We note here that there is one $\lambda_i$ associated to each observation, and they are assumed to factorize: $p(\lambda) = \prod_{i=1}^{N} p(\lambda_i)$. This choice of conjugate priors for variance estimation is similar to [Stirn and Knowles, 2020, Takahashi et al., 2018, Detlefsen et al., 2019]. Finally, we assume that, conditional on $I$ and $\theta$, we have marginal independence in $y$, i.e. $p(y|I, \theta) = \prod_{i=1}^{N} p(y_i|I_i, \theta_i)$.

## 4.2 VARIATIONAL APPROXIMATION

The integral equation (2) for the marginal likelihood is intractable and, thus, the posterior $p(\lambda, \theta|I, y)$ is too. We derive a lower bound on the log marginal likelihood to utilize variational inference [Blei et al., 2017]. We choose the variational approximation $q$ of the posterior $p(\theta, \lambda|I, y)$ as

$$q(\theta, \lambda) := p(\theta|\lambda, I) q(\lambda). \tag{7}$$

Here $p(\theta|\lambda, I)$ is given as before and $q(\lambda) := \prod_{i=1}^{N} \Gamma(\alpha_i, \beta(I_i))$. In our approximation, $\beta$ is a neural network: hence, we use amortized inference in a similar way to the VAE model [Kingma and Welling, 2014].

We derive our lower bound using Jensen's inequality

$$\log p(y, I) = \log \iint p(y, I, \theta, \lambda) \mathrm{d}\theta \mathrm{d}\lambda \tag{8}$$

$$\geq \iint \log \left( \frac{p(y, I, \theta, \lambda)}{q(\theta, \lambda)} \right) q(\theta, \lambda) \mathrm{d}\theta \mathrm{d}\lambda \tag{9}$$

$$= \iint \log \left( \frac{p(y|I, \theta) p(\lambda) p(I)}{q(\lambda)} \right) p(\theta|\lambda, I) q(\lambda) \mathrm{d}\theta \mathrm{d}\lambda$$

$$= \underbrace{\mathbb{E}_{q(\theta, \lambda)} \log p(y|I, \theta)}_{\textbf{classification loss}} + \log p(I) - \mathrm{KL}(q(\lambda)\|p(\lambda)).$$
$$\tag{10}$$

Thus, our evidence lower bound (ELBO) objective function (10), consists of two terms: a classification loss and a KL-term controlling the distance of the approximate posterior to the prior. During inference, we can disregard $\log p(I)$ as it does not depend on parameters of interest.

## 4.3 INFERENCE

The choice of variational posterior implies the following for the **classification loss**

$$\mathbb{E}_{q(\theta, \lambda)} \log p(y|I, \theta) \tag{11}$$

$$= \iint \log p(y|I, \theta) q(\theta, \lambda) \mathrm{d}\theta \mathrm{d}\lambda \tag{12}$$

$$= \iint \log p(y|I, \theta) p(\theta|\lambda, I) q(\lambda) \mathrm{d}\theta \mathrm{d}\lambda \tag{13}$$

$$= \int \log p(y|I, \theta) \int \mathcal{N}(\theta|\mu(I), \lambda) \Gamma(\lambda|\alpha, \beta(I)) \mathrm{d}\lambda \mathrm{d}\theta$$

$$= \int \log p(y|I, \theta) t_{2\alpha}(\theta|\mu(I)), \frac{\beta(I)}{\alpha}) \mathrm{d}\theta. \tag{14}$$

Here $t$ denotes a scaled and location-shifted Student's $t$-distribution with mean $\mu(I)$, scaling $\beta$, and $\alpha$ degrees of freedom. For clarity, the marginalized $q(\theta)$ is $t$-distributed. Here $p(y|I, \theta)$ is what previously was referred to as $p(y|T_\theta(I))$, i.e. the classifier conditioned the transformed $I$.

We approximate Eq. 14 using an unbiased estimate

$$\mathbb{E}_{q(\theta, \lambda)} \log p(y_i|I_i, \theta_i) \approx \frac{1}{S} \sum_{s=1}^{S} \log p(y_i|I_i, \theta_{i,s}), \tag{15}$$

$$\text{with } \theta_{i,s} \sim t_{2\alpha_i}(\cdot|\mu(I_i)), \frac{\beta(I_i)}{\alpha_i}) \tag{16}$$

and backpropagate through neural networks $\mu(I)$ and $\beta(I)$ with the reparametrization trick. In all experiments $\alpha_i = 1$.

Combining terms, the final ELBO we maximize becomes

$$\mathcal{L}_{p,q}(I, y) \approx \sum_{i=1}^{N} \frac{1}{S} \sum_{s=1}^{S} \log p(y_i|I_i, \theta_{i,s}) \\ - \mathrm{KL}\left( q(\lambda)\|p(\lambda) \right) + \mathrm{const}, \tag{17}$$

which is readily optimized using any gradient-based method. The KL-term is analytically tractable and differentiable between two gamma distributions.

In practice, following Higgins et al. [2016] we introduce a weight parameter $w$ to the KL-term. This requires us to tune $w$ but in turn makes the model robust to the choice of prior. We perform a grid-search on a validation set to find the optimal $w$. Alternatively, we could have done a grid search over $\beta_0$; instead we chose $\alpha_0 = \beta_0 = 1$ for all experiments. Similar to Kingma and Welling [2014], we often find it sufficient to draw only $S = 1$ samples during training. Note that our model naturally implies marginalization, and correspondingly data augmentation, at *test-time* as well as the usual training time. At test time, we draw $S = 10$ transformation samples.

# 5 EXPERIMENTS & RESULTS

Our model consists of two parts, the classifier $p(y|T_\theta(I))$ and the probabilistic localizer estimating the distribution over transformations. In the following experiments, we aim to disentangle our model's benefits for localization (Sec. 5.1), classification (Sec. 5.2) and calibration (Sec. 5.3).

The probabilistic localizer estimates $q(\theta) = t_2(\theta|\mu(I), \beta(I))$, i.e. in practice we implement a mean and a variance network, $\mu(I)$ and $\beta(I)$, respectively (see Fig. 2 for the architecture). We employ a small convolutional network (`Conv2d, Maxpool2d, ReLU, Conv2d, Maxpool2d, ReLU`) followed by two fully connected layers for both the localizer and classifier unless stated otherwise. The P-STN localizer has two heads; one for the mean and one for the variance. The number of parameters is stated in each experimental subsection. Unless stated otherwise, we keep the number of parameters constant, i.e. when adding a localization network we remove the extra parameters from the classifier for fair comparison.

Our model is implemented in PyTorch and experiments are run on 12 GB Nvidia Titan X GPUs. The code is available at `https://github.com/FrederikWarburg/pSTN-baselines`.

## 5.1 MARGINALIZING TRANSFORMATIONS IMPROVES LOCALIZATION ACCURACY

The appeal of STN models is that they are trained end-to-end, i.e. based only on labels for the downstream task, and not the transformations. This same property, however, is what makes the STN hard to fit. The only signal we obtain is through the supervised downstream task (i.e. the classification labels) and thus gradient information is sparse. We will now investigate whether estimating a posterior over transformations and marginalizing, i.e. 'getting to try multiple transformations', simplifies the task as suggested by Fig. 1.

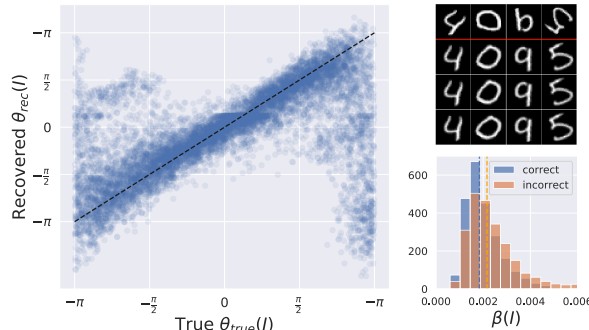

Figure 4: Rotated MNIST experiment. *Left panel:* Ground-truth transformation (rotation angles in radians) against recovered transformations (mean). *Top right:* Example images from the data set and samples from the P-STN localizer. The localizer learns to pose-normalize. *Bottom right:* Outputs of the variance network. When the transformation recovery is poor (the error $\varepsilon$ is above the median, in orange) the variances are slightly higher than when the localization works well (blue).

In order to disentangle the localization from the classification task, we construct the following experiments. We first train a CNN on a pose-normalized dataset (regular MNIST and Fashion MNIST). We then generate a new dataset by randomly sampling transformations $\theta_{\text{true}}$ and applying them to the MNIST images. Saving these transformations provides us with ground truth. We freeze the CNN weights and train STN and P-STN with this fixed classifier, effectively learning to recover and 'undo' the true transformations.

### 5.1.1 Rotated MNIST

From this data-generating process, we obtain a rotated version of the MNIST dataset (i.e. regular MNIST with ground-truth transformations given by rotation angles, $\theta_{\text{true}}(I) = r_{\text{true}}(I)$). See Fig. 4, top right panel for example data.

Our CNN classifier (28k weights) obtains $99.4\%$ test accuracy on MNIST and $41.2\%$ on rotated MNIST (frozen weights, no re-training). The STN and P-STN ($S = 10$ training samples, $w = 3 \cdot 10^{-5}$, same CNN classifier as before $+72$k params in the localizer) both learn to pose-normalize, i.e. to recover these transformations to a satisfactory degree. When training the localizers only (classifier weights remain frozen as described above), the STN test acc. is $76.13\%$, and $82.98\%$ for the P-STN. We compute the expected average transformation error on the $N = 10k$ rotated MNIST test images as

$$\varepsilon = \frac{1}{N} \sum_{i=1}^{N} \|\theta_{\text{true}}(I_i) - \mu(I_i)\| \mod \pi. \quad (18)$$

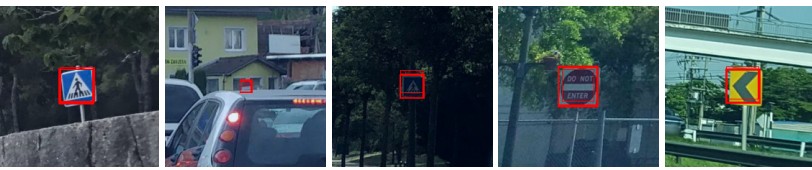

| | Acc. ↑ | NLL ↓ |
|---|---|---|
| CNN | 76.0 | 0.49 |
| STN | 90.6 | 0.31 |
| P-STN | **92.2** | **0.29** |

Figure 6: The P-STN learns to localize traffic signs in the challenging MTSD dataset. At test time, we sample 10 transformations as shown with the various bounding boxes overlaid the images. These learned variations improve the final classification.

Table 1: Accuracy (Acc.) and negative log-likelihood (NLL) for CNN, STN and P-STN.

We get $\varepsilon = 0.76$ for the STN and $\varepsilon = 0.59$ for the P-STN. The P-STN outperforms the STN, i.e. modeling uncertainty in the transformations helps in the localization task.

**Uncertainty.** The bottom right panel of Fig. 4 shows a histogram of $\beta(I)$, i.e. the localizer variance (or, correspondingly, the magnitude of augmentation) per image. In orange, we plot variances for images where pose-normalization is difficult (the transformation error $\varepsilon$ is larger than the median). In blue, we plot variances for images that are correctly pose-normalized (transformation error $\varepsilon$ smaller than the median). The poorly localized images are, on average, assigned 17% larger variances $\beta(I)$. The localizer uncertainty and thus the amount of data augmentation applied is somewhat meaningful, corresponding to the difficulty of the task.

### 5.1.2 Random placement FashionMNIST

We repeat a similar experiment on the slightly more challenging FashionMNIST dataset [Xiao et al., 2017] . The CNN baseline accuracy is $90.63\%$ (same model as above with 28k parameters). We then randomly sample an $x$ and $y$ coordinate and place the FashionMNIST accordingly on a black background, after downscaling it by $50\%$. No rotation is applied, i.e. $\theta_{\text{true}} = [0, 0.5, t^x_{\text{true}}, t^y_{\text{true}}]$.

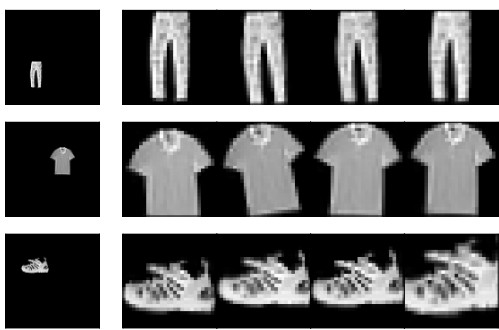

Figure 7: Random Placement Fashion MNIST. Input images (left) and transformed samples $T_{\theta_s}(I)$ as learned by the P-STN. The P-STN learns to correctly pose-normalize and zoom into the relevant part of the image. The samples look like plausible candidates for a data augmentation scheme. We will explore this in Sec. 5.2.

Like in the previous experiment, both localizers successfully recover $\theta_{\text{true}}$, with the P-STN ($S = 10$ training samples, $w = 3e{-}05$, same classifier as before +193k weights in the localizer) doing slightly better than its deterministic counterpart: test accuracies are $84.99\%$ and $84.41\%$, respectively. Inspecting the transformation posterior and the resulting samples $T_{\theta_s}(I_i)$, we find that these look visually pleasing, and, as hypothesized, might be promising candidates for a data augmentation scheme. We will explore this in Sec. 5.2.

### 5.1.3 Mapillary street signs

Detection and classification of objects in images have many applications, e.g. for autonomous vehicles, detecting traffic signs is crucial. We compare a top-performing classifier, an STN and our P-STN on the challenging Mapillary Traffic Sign Dataset (MTSD) [Ertler et al., 2019].

To focus this comparison, we select images that contain only one traffic sign. We obtain this subset by selecting all bounding boxes that do not intersect with other bounding boxes plus a margin of 150 px to each side. We further select the ten most common classes from this subset. This gives us a training set of 4698 images and a test set of 500 images. Figure 6 shows example images from the chosen subset.

Our classifier is a ResNet18 pre-trained on ImageNet, where we replace the last fully connected layer. We use the same ResNet for the localizers in the STN and P-STN, where we similarly replace the last layer. As before, we wish to study the behavior of the localizers. Therefore, we again start by training a classifier on the ground-truth bounding boxes. We then initialize the classifier module of the STN and P-STN with this pre-trained classifier and freeze the weights of the classifier. We train the localizers of the STN and P-STN for 60 epochs with learning rate $10^{-4}$ and kl weight $w = 10^{-7}$. Figure 6 shows that the P-STN learns to localize the traffic signs. At test time, we sample 10 transformations illustrated by the multiple overlaying bounding boxes.

Table 1 shows that both the STN and P-STN clearly outperform the baseline classifier when trained on the full images. Even though the STN and P-STN have exactly the same classifier, the P-STN achieves better performance because of the ensemble of classified transformations.

|              | MNIST30        | MNIST100       | MNIST1000      | MNIST3000          | MNIST10000       |
| ------------ | -------------- | -------------- | -------------- | ------------------ | ---------------- |
| *CNN*        | $70.12 \pm 2.46$ | $87.29 \pm 0.58$ | $95.80 \pm 0.33$ | $\mathbf{97.48} \pm 0.21$ | $\mathbf{97.82} \pm 0.34$ - |
| *affine STN* | $69.26 \pm 4.53$ | $82.16 \pm 2.30$ | $92.05 \pm 0.58$ | $94.71 \pm 0.22$ | $96.96 \pm 0.20$ |
| *affine P-STN* | $\mathbf{81.00} \pm 3.92$ | $\mathbf{92.70} \pm 0.74$ | $\mathbf{96.62} \pm 0.58$ | $97.33 \pm 0.17$ | $97.63 \pm 0.23$ |
| *optimal w*  | 0.001          | 0.0003         | 0.0001         | 0.00003            | 0.00001          |

Table 2: The performance of a CNN, STN and P-STN on differently sized MNIST datasets. Bold numbers indicate that a model is significantly better than the runner up under a two sample t-test at $p = 0.05$.

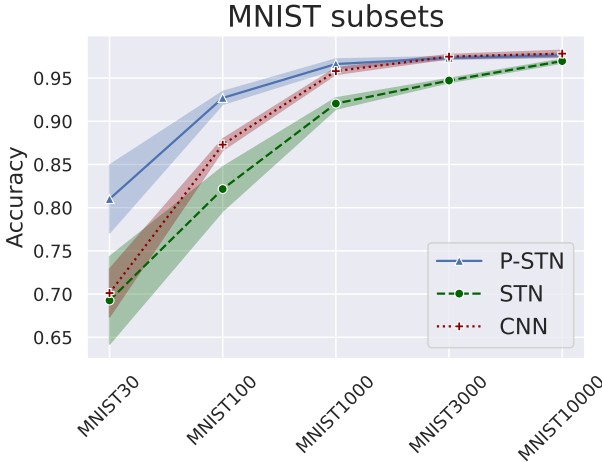

Figure 8: Performances of P-STN, STN and CNN on MNIST subsets (mean $\pm$ one STD across five folds).

## 5.2 MARGINALIZING TRANSFORMATIONS IMPROVES CLASSIFICATION ACCURACY

We have argued that marginalizing transformations via samples corresponds to learned, localized data augmentations (the samples $T_{\theta_s}(I)$). We will now investigate whether these augmentations are indeed helpful in the downstream task, i.e. whether they improve classification performance.

### 5.2.1 MNIST and subsets

We compare the performance of our P-STN against a standard convolutional neural network (CNN) and a regular STN on MNIST. The standard MNIST images are centered and pose-normalized, so the localization task is easy. Improved classifier performance can thus be viewed as an indicator for having learned a useful data augmentation scheme.

Data augmentation is particularly important when training data is scarce, so we evaluate the models on small subsets of MNIST: MNIST30 contains 30 images (i.e. 3 per class), MNIST100, MNIST1000, MNIST3000 and MNIST10000. STN and P-STN parameterize affine transformations, i.e. the learned $\theta$ is interpreted as the full affine matrix as described in Sec. 3. All models have roughly 28k parameters,

architecture as described at the top of Sec. 5. We use the Adam optimizer with weight decay 0.01 and the default parameters of its PyTorch implementation. The images are color-normalized. We repeat the experiment 5 times, each time with a different $k$-image subset of the MNIST dataset, and we report $\pm$ one standard deviation in tables and error bars. From Table 2 and Fig. 8, we see that the P-STN outperforms both the STN and CNN on the small dataset sizes. For the larger datasets, the differences vanish. This supports our hypothesis: data augmentation is especially useful when data is a limited resource. This intuition is also supported by the optimal KL-weights (Table 2, bottom row) that we determine via grid search on validation data. For smaller datasets, larger $w$ and thus more regularization towards the variance prior (away from 0) are beneficial.

The fact that the STN performs less well than the standard CNN on this data set might be explained by the fact that the images are already nearly perfectly pose-normalized, and wrong transformations can be detrimental.

### 5.2.2 UCR time-series dataset

For some data modalities, such as time-series, it is not trivial to craft a useful data augmentation scheme. In this experiment, we show that the P-STN can learn a useful, non-trivial data augmentation scheme that increases performance compared to a standard STN on time-series data. The UCR dataset [Dau et al., 2018] is composed of 108 smaller datasets, where each dataset contains univariate time-series. The FordA dataset, for example, contains measurements of engine noise over time and the goal is to classify whether or not the car is faulty. We select 5 of those subsets, each large enough to divide into training and validation sets (75/25%), which we use to find the optimal $w$ via grid-search; those are $[0.0001, 1e - 05, 0.001, 0.0, 0.0001]$. We draw $S = 10$ training samples. The test-set is pre-defined by the dataset curators. Learning rate and optimizer are the same as in Sec. 5.2.1, but we do not perform normalization. All models have approximately one million parameters. Table 3 shows that the P-STN achieves higher mean accuracy than both the STN and the CNN, indicating that we can automatically learn a useful data augmentation scheme for time-series.

We verify this qualitatively in Fig. 10, which shows an

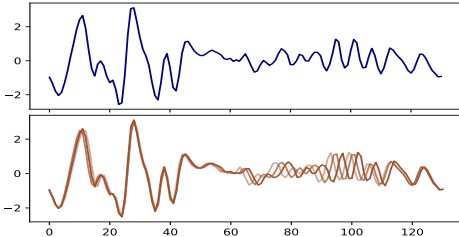

Figure 10: Examples of augmentations for a time-series from the FaceAll dataset. The top plot shows the original time-series and the bottom plot shows three augmented versions of the time-series.

| | CNN | STN | P-STN |
|---|---|---|---|
| FaceAll | $80.83 \pm 0.62$ | $82.28 \pm 0.42$ | $\mathbf{84.31} \pm 0.75$ |
| TwoPatterns | $97.92 \pm 0.53$ | $99.79 \pm 0.04$ | $\mathbf{99.96} \pm 0.04$ |
| wafer | $\mathbf{99.63} \pm 0.05$ | $99.18 \pm 0.17$ - | $98.86 \pm 0.20$ |
| uWaveGestureLib.* | $74.15 \pm 1.27$ | $79.77 \pm 0.42$ - | $\mathbf{81.13} \pm 0.46$ |
| PhalangesOutlC.** | $79.88 \pm 1.32$ | $\mathbf{82.26} \pm 0.98$ | $81.66 \pm 0.59$ |
| Mean | 86.48 | 88.65 | **89.18** |

Table 3: Accuracies on a subset of the UCR timeseries dataset (full dataset names are *uWaveGestureLibrary and **PhalangesOutlinesCorrect). $\pm 1$ STD is reported after 5 repetitions. Bold numbers indicate that a model is significantly better than the runner up under a two sample t-test at $p = 0.05$.

## MNIST100 Calibration

Figure 11: Calibration plots for CNN, STN and two P-STN models. One with KL-weight yielding optimal performance ($w = 0.0003$) and one with KL-weight yielding optimal calibration ($w = 0.0001$). Both P-STN models are better calibrated than CNN and STN.

example of the learned data augmentation. We see that the model does not simply apply a global transformation, but learns to augment the time-series more in some intervals, such as in $[60; 110]$, and augment the time-series less in other intervals, such as in $[0; 50]$.

### 5.3 MARGINALIZING TRANSFORMATIONS IMPROVES CALIBRATION

In Sec. 5.1, we have seen that harder images on average have larger transformation uncertainties. We now investigate whether those meaningful localization uncertainties translate into meaningful uncertainties downstream, i.e. in the calibration of our classifier. At test-time, we evaluate

$$p(y|I) = \int p(y|I, \theta)q(\theta)\mathrm{d}\theta \approx \frac{1}{S}\sum_{s=1}^{S} p(y|T_{\theta_s}(I)). \quad (19)$$

We will investigate how well the uncertainty in this distribution matches the quality of predictions. Fig. 11 shows a calibration plot for the MNIST100 subset classification task from Sec. 5.2.1 for the CNN, STN and P-STN for two different $w$-parameters; $w = 0.0003$ yields the best performance (reported in Table 2) and $w = 0.0001$ yields the best calibration. The expected calibration errors [Guo et al., 2017, Küppers et al., 2020, 2021] are CNN: $0.0743 \pm 0.0094$, STN: $0.1160 \pm 0.0205$, P-STN, $w = 0.0003$ (optimal performance model): $0.0567 \pm 0.0065$, P-STN, $w = 0.0001$ (optimal calibration model): $\mathbf{0.0271} \pm 0.0088$. We report the mean over 5 folds, $\pm$ one STD. The P-STN significantly improves calibration in the downstream classification task.

### 5.4 A TYPICAL FAILURE MODE IN STNS

STNs are trained end-to-end, and with only label information available. Thus, the aim is to learn the optimal transformation for solving the downstream task. Depending on the complexity of the downstream task and the classification model, it might not be necessary to transform the input at all, i.e. it might be possible to solve the downstream task on the original input image. Indeed, this is a failure mode we observe in practice — often, the localizer simply learns the identity transform while the classifier learns to classify the non-transformed image. Using more complex classifier architectures makes the STN more prone to this failure mode. This has been observed by other authors [Finnveden et al., 2021], and we investigate the problem in the experiment in Fig. 12. We start by training differently-sized neural networks on MNIST (black, one layer on the $x$-axis is [Linear, ReLU, Dropout]). We compare the performance of this model with (P-)STN models trained on rotated MNIST, test accuracies are plotted in the left panel of the figure. If the localization task is performed perfectly, the (P-)STN models should be able to recover the accuracy on the original, non-rotated dataset. In the right panel, we plot the variance of the (mean) transformations learned by the (P-)STN models. Values close to 0 indicate that the localizer does not transform the image, i.e. it learns the

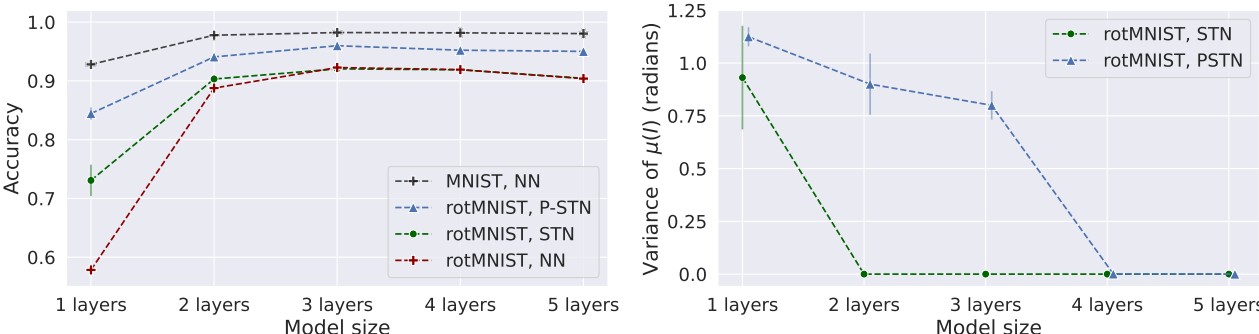

Figure 12: *Left:* Test accuracies for standard NN and (P-)STNs of different depths trained on rotated MNIST, as well as NN baseline on original MNIST (black). The STN (green) model does not usually recover the original images and thus behaves more like a standard NN (red) in most runs. P-STN (blue) un-transforms at least some of the rotations and is closer in accuracy to the NN on original MNIST (black). *Right:* The variance of the learned transformations as a function of model depth. The STN learns the identity for deeper downstream models (this is consistent with the test accuracies we see on the left). P-STN learns to un-transform better, at least when the classifier is simple. For bigger classifiers it predicts the identity transform as well, but performs relatively well nonetheless (see left panel). We report medians $\pm 1$ median absolute deviation over 5 folds.

identity transform. Larger values indicate that the localizer learns transformations. Median results are reported over 5 runs, error bars correspond to one mean absolute deviation. As hypothesized, for larger classifiers the localizers do not transform the images. Due to the increased capacity of the model, we nonetheless achieve decent classification accuracies (left panel). The P-STN learns to localize the rotated images somewhat successfully (large variance in the right panel, and high accuracy on the left) for smaller classifiers. The STN does not localize the images as well, most runs behave like the standard NN on rotMNIST (red), predicting identity transformations only. We conclude that, thanks to it 'trying out multiple transformations', the P-STN avoids this failure mode to an extent. We also note that this property, while useful, is somewhat orthogonal to our interest in this work, and we have avoided the failure mode in the experiments of Sec. 5.1 by considering models with *fixed*, pre-trained classifiers.

## 6 CONCLUSION

We have introduced a probabilistic extension to the spatial transformer network (STN) [Jaderberg et al., 2015]. Our work took motivation from the empirical observation that the STN is often brittle to train, as a poorly predicted transformation may prevent the model from getting any gradient signal, resulting in divergent optimization. Our probabilistic STN (P-STN) instead approximates the posterior distribution of transformations using amortized variational inference, and marginalizes accordingly. As is common, marginalization improves the robustness of the model.

Empirically, we note the following advantages of the prob-

abilistic formulation over the deterministic. Firstly, the performance of the localization network is improved, since the Monte Carlo marginalization effectively amounts to trying many different transformations. Secondly, the probabilistic formulation improves the overall model performance, since the sampled transformations act as data augmentation both during training and during testing. The resulting ensemble of predictions is more accurate and better calibrated than common classifiers as well as the original spatial transformer.

## Acknowledgements

MJ was supported by a research grant from the Carlsberg Foundation (CF20-0370). SH was supported by research grants (15334, 42062) from VILLUM FONDEN. This project has also received funding from the European Research Council (ERC) under the European Union's Horizon 2020 research and innovation programme (grant agreement No. 757360). This work was funded in part by the Novo Nordisk Foundation through the Center for Basic Machine Learning Research in Life Science (NNF20OC0062606).

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
