# OpenReview forum: "Probabilistic Spatial Transformer Networks"
_auai.org/UAI/2022/Conference — UAI 2022 Poster_

### Official Review · Reviewer_dQcg · 2022-04-08

**Q2(1) Originality/Novelty:** 3
**Q2(2) Significance/Impact:** 3
**Q2(3) Correctness/Technical Quality:** 3
**Q2(6) Clarity Of Writing:** 3
**Q6 Overall Score:** 7
**Q8 Confidence In Your Score:** 4

**Q1 Summary And Contributions:**

•	STNs are difficult to train correctly due to extreme sensitivity to error.
•	The paper proposes a stochastic method to make STNs more robust.
•	In this paper, the localizer is sampled for many transformations instead of the estimated maximum likelihood transformation.
•	This method is most obviously applicable to the image classification pipeline but can be used in other image processing tasks or non-image tasks like time series.


**Q2 Assessment Of The Paper:**

More detailed information regarding each of these aspects is given below:

**Q2(4) Quality Of Experiments (Optional):**

3: Good: The experimental evaluation is adequate, and the results convincingly support the main claims.

**Q2(5) Reproducibility:**

4: Excellent: Key resources (e.g., proofs, code, data) are available and key details (e.g., proof sketches, experimental setup) are comprehensively described for competent researchers to confidently and easily reproduce the main results.

**Q3 Main Strengths:**

•	Code is promised alongside this paper.
•	Experimental coverage goes above and beyond. Common benchmarks, realistic data, and data from a secondary application domain are used.
•	Algorithm tested against CNN and unmodified STN.
•	Improvements are shown in most cases.
•	Most of the text is very clear and readable.
•	The contribution may be a small step, but it is mathematically interesting and sound.


**Q4 Main Weakness:**

•	Main criticism: across most experiments, improvements seem tenuous. Likely not statistically significant.
•	Figures and tables are a bit hard to follow. More complete labeling and captions (including units) would make them easier to read.


**Q5 Detailed Comments To The Authors:**

The paper is well-written. It presents an interesting idea (looks small but interesting) and defends it with reasonable experimental results. The authors may look into other sampling approaches which can probably improve their algorithm even more. Overall, a good paper.

**Q7 Justification For Your Score:**

The paper is solidly written and scientifically sound. Code is promised and hyperparameters are given. If results were stronger and significance was shown, would be at least an 8.

**Q9 Complying With Reviewing Instructions:**

1: Yes.

---

### Official Review · Reviewer_pswS · 2022-04-10

**Q2(1) Originality/Novelty:** 3
**Q2(2) Significance/Impact:** 3
**Q2(3) Correctness/Technical Quality:** 3
**Q2(6) Clarity Of Writing:** 3
**Q6 Overall Score:** 6
**Q8 Confidence In Your Score:** 4

**Q1 Summary And Contributions:**

This paper proposes a stochastic spatial transformation network for stabilized and probabilistic data augmentation. This transformation is built upon a graphical model and the augmentation is treated as the latent variable. The whole framework is optimized by maximizing a tractable variational lower bound. The learned transformation network allows stochastic localized augmentation for robust classification performance and calibrated predictive model.

**Q2 Assessment Of The Paper:**

More detailed information regarding each of these aspects is given below:

**Q2(4) Quality Of Experiments (Optional):**

2: Fair: The experimental evaluation is weak: important baselines are missing, or the results do not adequately support the main claims.

**Q2(5) Reproducibility:**

2: Fair: Key resources (e.g., proofs, code, data) are unavailable but key details (e.g., proof sketches, experimental setup) are sufficiently well-described for an expert to confidently reproduce the main results.

**Q3 Main Strengths:**

The proposed method enjoys both pleasing theoretic and empirical results. First, the data augmentation is presented with a graphical model where the augmented data is the latent variable. Then, to learn a probabilistic transformation, it resorts to Bayesian learning and induces a tractable lower bound for optimization, which is technically sound. Moreover, empirical results on various MNIST dataset show its superior performance.

**Q4 Main Weakness:**

I think the main weakness lies in the experiment part. Only small datasets like MNIST/Fashion MNIST are used for evaluation. I wonder if the proposed method can outperform the baseline methods on larger datasets like CIFAR10 and CIFAR100.

**Q5 Detailed Comments To The Authors:**

The proposed method enjoys both pleasing theoretic and empirical results. First, the data augmentation is presented with a graphical model where the augmented data is the latent variable. Then, to learn a probabilistic transformation, it resorts to Bayesian learning and induces a tractable lower bound for optimization, which is technically sound. Moreover, empirical results on various MNIST dataset show its superior performance.

However, only small datasets like MNIST/Fashion MNIST are used for evaluation. I wonder if the proposed method can outperform the baseline methods on larger datasets like CIFAR10 and CIFAR100.


**Q7 Justification For Your Score:**

The proposed method enjoys both pleasing theoretic and empirical results.

**Q9 Complying With Reviewing Instructions:**

1: Yes.

---

### Official Review · Reviewer_Q3n1 · 2022-04-12

**Q2(1) Originality/Novelty:** 3
**Q2(2) Significance/Impact:** 3
**Q2(3) Correctness/Technical Quality:** 3
**Q2(6) Clarity Of Writing:** 4
**Q6 Overall Score:** 6
**Q8 Confidence In Your Score:** 4

**Q1 Summary And Contributions:**

This paper formulates the problem of applying spatial transformations to images from the probabilistic perspective. Marginalizing over the transformations successfully solves the hard-to-train problem of the original STNs and further provides useful data augmentations that enhance the performance of downstream tasks. A variational amortized inference algorithm is proposed to learn the model in an automatic fashion.


**Q2 Assessment Of The Paper:**

More detailed information regarding each of these aspects is given below:

**Q2(4) Quality Of Experiments (Optional):**

3: Good: The experimental evaluation is adequate, and the results convincingly support the main claims.

**Q2(5) Reproducibility:**

3: Good: Key resources (e.g., proofs, code, data) are available and key details (e.g., proofs, experimental setup) are sufficiently well-described for competent researchers to confidently reproduce the main results.

**Q3 Main Strengths:**

1. The paper recast the problem of STN from a Bayesian perspective and derives a variational inference algorithm to solve this problem.

2. The paper is well motivated and well written.

3. Experimental results show the strength of the proposed method in both image and time series data.


**Q4 Main Weakness:**

My main concern lies in how to show that P-STN truly solves the issues with the original STN. The original STN is hard to train and the spatial transformations may harm the performance when the downstream CNN is deep. It is better to find an example and show that P-STN performs much better than STN when it is hard to train STN. Or P-STN still works when the CNN is relatively deep.



**Q5 Detailed Comments To The Authors:**

1. The proposed method is also related to ensemble learning, such as bagging. It’s better to mention these works and point out the difference.

2. It’s better if the authors can provide some theoretical support that validate the superiority of P-STN over STN.


**Q7 Justification For Your Score:**

The formulation of the problem and the algorithm is novel. The weaknesses do not affect the novelty of the paper. But I suggest the authors consider my concerns and detailed comments in the revision.

**Q9 Complying With Reviewing Instructions:**

1: Yes.

---

### Official Review · Reviewer_Vryc · 2022-04-12

**Q2(1) Originality/Novelty:** 2
**Q2(2) Significance/Impact:** 3
**Q2(3) Correctness/Technical Quality:** 4
**Q2(6) Clarity Of Writing:** 3
**Q6 Overall Score:** 6
**Q8 Confidence In Your Score:** 4

**Q1 Summary And Contributions:**

Spatial Transformer Networks (STN) deduce image transformations that help in downstream tasks. This paper provides a probabilistic extension of STNs that marginalizes over multiple transformations making the model robust. Moreover the probabilistic model acts as a data augmenter. The authors provide variational approx inference for their proposed model. Experiments on both synthetic and real world data shows that marginalizing over transformations provides a robust model that outperforms STNs.

**Q2 Assessment Of The Paper:**

More detailed information regarding each of these aspects is given below:

**Q2(4) Quality Of Experiments (Optional):**

4: Excellent: The experimental evaluation is comprehensive and the results are compelling.

**Q2(5) Reproducibility:**

3: Good: Key resources (e.g., proofs, code, data) are available and key details (e.g., proofs, experimental setup) are sufficiently well-described for competent researchers to confidently reproduce the main results.

**Q3 Main Strengths:**

-- The paper proposes probabilistic extension of STNs. The proposed model is a simple extension of STNs

-- experiments on Mapillary Traffic Sign Dataset (MTSD) clearly shows that marginalizing over the transformations improves accuracy over STNs

-- In the low data regime, the proposed model seems to outperform STNs by a lot (section 5.2)

-- Moreover they empirically show that both the best performing model and the best calibrated model are pSTNs

**Q4 Main Weakness:**

-- The data augmentation process is not very clear (section 5.2)

-- Even though the proposed extension to STNs is only somewhat novel, the detailed experiments section makes up for it.

**Q5 Detailed Comments To The Authors:**

I would have loved to see some analysis of the increase in computational time as compared to STNs.

**Q7 Justification For Your Score:**

please see Q2 and Q3

**Q9 Complying With Reviewing Instructions:**

1: Yes.

---

### Decision · Program_Chairs · 2022-05-15

**Decision:**

Accept (Poster)

**Comment:**

Meta Review: This paper proposes a stochastic spatial transformation network for stabilized and probabilistic data augmentation. Reviewers appreciate the novelty and theoretical justification, and the empirical evaluations are comprehensive and valid. Reviewer 3 suggests more experiments on larger datasets, which the authors are encougraged to include in the final version. In all, this is a good paper and the meta-reviewer recommends acceptance.